# The Analysis of the Relationship between the Quality of Life Level and Expectations of Patients with Cardiovascular Diseases under the Home Care of Primary Care Nurses

**DOI:** 10.3390/ijerph19063300

**Published:** 2022-03-11

**Authors:** Elżbieta Szlenk-Czyczerska, Marika Guzek, Dorota Emilia Bielska, Anna Ławnik, Piotr Polański, Donata Kurpas

**Affiliations:** 1Institute of Health Sciences, University of Opole, 68 Katowicka Street, 45-060 Opole, Poland; 2Medical and Diagnostic Centre (MDC), 9 Niklowa Street, 08-110 Siedlce, Poland; marika.guzek@centrum.med.pl; 3Department of Family Medicine, Medical University of Białystok, 1 J. Kilińskiego Street, 15-089 Białystok, Poland; d.bielska1@wp.pl; 4Faculty of Health Sciences, Pope John Paul II of Higher State School Education, 95/97 Sidorska Street, 21-500 Biala Podlaska, Poland; lawnikania@gmail.com; 5Family Physician’s Practice, Non-Public Healthcare Center, 4 Nad Potokiem Street, 58-350 Mieroszow, Poland; p.polanski@wp.pl; 6Department of Family Medicine, Wrocław Medical University, 1 Syrokomli Street, 51-141 Wrocław, Poland; dkurpas@hotmail.com

**Keywords:** chronic cardiovascular disease, patients, expectations, quality of life level

## Abstract

The World Health Organization defines quality of life as a person’s perception of his or her life situation in relation to the culture and value system in which he or she lives, in relation to and with respect to his or her functioning assumptions, expectations, and standards set by environmental conditions. Meeting the expectations of patients with CVD is one of the factors that positively influences their health status and leads to better diagnostic and treatment outcomes. The aim of this study was to answer three main questions related to patients with chronic cardiovascular disease: (1) What is their quality of life? (2) Are patients’ expectations about the quality of care provided by primary health care physicians/nurses met (and at what level)? (3) Is there a correlation between patients’ quality of life and their expectations of primary health care physicians/nurses? The study involved 193 Polish CVD patients who were cared for at home by a family nurse practitioner working in primary health care facilities. Data were collected from March 2016 to January 2017. The WHOQOL-BREF Quality of Life Questionnaire and the Author Interview Questionnaire were used for the study. Data analysis was based on the Spearman correlation coefficient test. There was a statistically significant association between patients’ expectations of the physician regarding information about the course of the disease and quality of life in the following domains: environmental r = 0.20, *p* = 0.006, psychological: r = 0.18, *p* = 0.015, physical: r = 0.18, *p* = 0.013, and social: r = 0.16, *p* = 0.025. Patients who did not expect the nurse to be courteous, understanding, or interested were found to have higher quality of life scores in psychological (r = −0.17, *p* = 0.023) and physical (r = −0.15, *p* = 0.044) domains. There was a statistically significant relationship between expectations of care from nurses regarding intimacy during care activities and the level of satisfaction with one’s own health (r = −0.15, *p* = 0.038) and quality of life (r = −0.14, *p* = 0.045), as well as quality of life in the domains of physical (r = 0.21, *p* = 0.004), social (r = 0.19, *p* = 0.010), and psychological (r = 0.16, *p* = 0.024). There is a need to define the expectations of patients with chronic cardiovascular disease in primary care, as lack of expectations of a physician/nurse continues to be associated with lower quality of life in all domains.

## 1. Introduction

Chronic diseases usually bring negative consequences that affect various aspects of a patient’s life. The consequences may be temporary or permanent and, in most cases, result in the need for changes in previous lifestyle and long-term treatments. The assessment of a patient’s clinical efficacy and functional well-being in chronic disease is extremely important as it is a major cognitive and practical factor in the selection of alternative therapies [1,2,3].

Due to the wide prevalence of cardiovascular disease, this remains one of the most studied areas in quality of life (QoL) assessment. Therefore, it is of great interest to assess the physical, psychological, and social impact of disease on individuals’ lives. Numerous studies have shown that QoL assessment in this patient population is as important as physical, laboratory, or clinical test results. More importantly, quality of life assessment influences the effectiveness of medical treatment by making treatment choices easier and more acceptable to the patient. It is also considered an indicator of the effectiveness of current social and political support systems [2,3,4]. The aspects of quality of life considered include physical state (mobility and independence), emotional state (depressive symptoms, anxiety, anger, mood swings, feelings of shame, helplessness, and expectations for the future), social relationships (social, sexual, and family activities, and satisfaction with married life), economic status (income and employment), intellectual abilities (memory, ability to concentrate, and ability to learn), and self-perception of one’s health status (self-assessment of severity of symptoms and degree of disability) [4,5,6,7,8].

The primary goal of chronic cardiovascular disease management is to improve patient functioning and reduce treatment costs by reducing unnecessary health care interventions [9,10,11,12]. As most studies have shown, both goals can be achieved by improving quality of life [13]. It has been reported that patients who use health care services more frequently have lower quality of life in the physical, psychological, and social domains [13,14]. It has also been found that the higher the quality of life of patients, the less primary health care procedures are used [15,16].

However, the problem of patients’ satisfaction with nursing and medical care of patients with chronic CVD is still a rare aspect in various analyses. Meeting their expectations is one of the factors that positively influences their health status and leads to better diagnostic and treatment outcomes [17]. Therefore, patients’ evaluation of medical services forms the basis for improving the quality of primary health care and can provide information about the development goals and the need for medical care in specific patient groups [15]. When patients’ expectations of a health care system are properly identified and met, it leads to higher satisfaction with the physician–patient contact. The level of satisfaction correlates positively with improvement in a patient’s clinical condition. Health care where the patient is the primary caregiver has been shown to be associated with lower mortality rates and lower risk of hospital complications [15,18,19].

The global burden of cardiovascular disease, the expected increase in morbidity, and demographic changes and their consequences are prompting many countries to update their health services to improve their effectiveness and reduce health inequalities. In Poland, too, the current assumptions and goals of public health programs assume a systematic increase in the number of patients with a chronic CVD diagnosis. As a result, the demand for health care services will also increase. Therefore, there is a need to redefine a more targeted and effective model of home care for patients with chronic CVD that can guide the planning of a more professional and interdisciplinary system. Assessing the relationship between CVD patients’ quality of life and their expectations of the health care system can potentially contribute to the development of a support system provided by skilled, interdisciplinary therapeutic teams.

With this in mind, the study aimed to analyze the quality of life of CVD patients and their expectations from a PHC physician/nurse and to determine if there is a correlation between patients’ quality of life and their expectations.

## 2. Materials and Methods

### 2.1. Study Design

This study is a cross-sectional observational study and is part of a larger study to identify indicators that determine the effectiveness of home care for patients with chronic CVDs. The study involved 350 patients with CVDs. To define specific indicators of home care, 193 patients remained at home under the care of family nurses, while 157 patients visited their primary care physicians for follow-up appointments. The study also included 161 caregivers of patients who received home care from family nurses. This article presents a partial analysis of the results of this study, which addresses the variables that affect the quality of life of CVD patients cared for at home by a family nurse practitioner and the expectations these patients have of the health care system (a PHC physician and a family nurse practitioner).

### 2.2. Sample

The study was performed on Polish CVD patients. These patients were cared for at home by a family nurse working in primary care facilities in Opole, Dolnośląskie, Mazovia, Lubelskie, and Podlaskie voivodeships. Eight primary care facilities participated in the study. Patients were encouraged to participate in the study by a family nurse practitioner during scheduled home visits. Respondents completed the questionnaires in their homes. Patients each received one set of questionnaires, and nurses completed an additional questionnaire about the patient (i.e., paired questionnaires related to the same patient). Data were collected from March 2016 to January 2017.

In our study, we used a nonprobabilistic sampling method (purposive sampling). Two hundred CVD patients who were cared for at home by family nurses were invited to participate in the survey. The final sample of participants was determined based on their time availability. Ultimately, 193 patients participated in the survey. The criteria for inclusion in the study were as follows: 18 years of age or older, confirmation of chronic CVD at least 12 months before the study, and residence under home care of a family nurse practitioner defining chronic CVD and determined based on a primary health care history. The exclusion criteria (disqualification by the family nurse practitioner) were cognitive impairment and other severe mental illness and/or other difficulties that prevented active participation in the study.

Participation in the study was voluntary and anonymous. All participants were informed of the aims and methods of the study and had the opportunity to withdraw their participation at any stage of the study.

### 2.3. Ethical Aspects

The study was approved by the Bioethical Committee of the Medical University in Wroclaw (No. KB-86/2016).

### 2.4. Variables and Data Collection

The quality of life of patients with CVD was assessed using the Quality of Life Short Form (WHOQOL-BREF) scale standardized by the World Health Organization (WHO, Geneva, Switzerland). It consists of 26 questions related to different aspects of life, such as physical, psychological, social relationships, and environment. It also includes two independently evaluated questions related to individual perceptions of (1) quality of life and (2) health status. The physical domain measures activities of daily living, dependence on medications or treatments, energy, fatigue, mobility, pain, discomfort, rest, sleep, and work readiness. The psychological domain allows respondents to self-report their physical well-being, positive and negative feelings, religion, spirituality, beliefs, cognitive abilities, learning, memory, and concentration. The social relations domain includes aspects such as personal relationships, sexual activity, and social support, while the environment domain assesses financial satisfaction, feeling free, feeling safe, quality of life and access to health care, housing, access to information, relaxation, and ability to pursue one’s interests. Finally, the environment section examines pollution, noise, traffic, climate, and transportation. Respondents rate each aspect on a 5-point scale (very poor, poor, neutral, good, and very good). The rating of the domains reflects individual perceptions of the QOL domains and has a positive direction—the higher the rating, the higher the QOL. The total score for each domain is calculated by counting the average of all items in each domain. The reliability of the Polish version of WHOQOL-BREF was checked using the α-Cronbach coefficient, which was 0.81 in physical, 0.78 psychological, 0.69 in social relations, and 0.77 environmental domains. The internal consistency of the whole questionnaire was 0.90 [20,21].

To capture key sociodemographic characteristics of CVD patients, the authors used a self-administered interview questionnaire that collected data on age, gender, education level, marital status, place of residence, and material status. It also included questions about patients’ expectations of a PHC physician or family nurse practitioner. In addition, the questionnaire included two questions about the patient’s sense of safety and improvement in well-being associated with awareness of the presence and visits of a family nurse practitioner.

### 2.5. Data Analysis

The results of the study were statistically analyzed using the statistical package R (version 3.4.0).

For quantitative variables, the arithmetic mean, standard deviation, first quartile (Q. 25%), median (Q. 50%), third quartile (Q. 75%), minimum, and maximum were calculated. For nominal variables, the frequency (i.e., percentage) was determined. The Shapiro–Wilk test showed that only some quantitative variables (i.e., staying in a relationship, WHOQOL-BREF—physical, psychological, and environmental) had a normal distribution. The other variables included deviated from the normal distribution. The chi-square test was used to assess the qualitative variables.

The Wilcoxon test and the post-hoc Quade test were used to assess quality of life according to the WHOQOL-BREF. The first test tests the null hypothesis for related samples when the distribution of variables in both related samples is the same, compared to the alternative that they are different. The second is a multiple comparisons test that calculates the significance of the difference for each pair of variables (domains). It can be performed only for the group of subjects for which all values for the variables were recorded. The test does not require that the distribution of the variables is normal (*p* = 0 means that *p* < 0.001). A significance level of 0.05 was assumed in the study.

The relationship between the quality of life of patients with CVD and their expectations of primary care physicians/nurses was analyzed using Spearman’s rank correlation coefficient, which does not assume a normal distribution of the variables. The null hypothesis (H0) was tested, with Spearman’s rank correlation coefficient equal to 0. The alternative hypothesis was that the correlation coefficient was different from 0. The null hypothesis (H0) was rejected if the *p* value < was 0.05 (α = 0.05).

## 3. Results

### 3.1. Sociodemographic Data of Patients with CVD

The sample group of CVD patients was mainly dominated by women, who accounted for 68.2% (*n* = 131). The median age in the group was 74 (*p* < 0.001). It was found that the median progression of a CVD or CVDs was 10 years (*p* < 0.001). Analysis of the variable “being in a relationship” revealed that half of the patients were in a committed relationship and the other half were not, namely 52.7%, *n* = 98 and 47.3%, *n* = 88 (*p* = 0.463). Respondents were more often urban residents—60.6%, *n* = 117—than village residents—39.4%, *n* = 76 (*p* = 0.003). Most of them had primary education—31.2%, *n* = 59; vocational education—25.4%, *n* = 48, or secondary education—25.5%, *n* = 48 (*p* < 0.001). The majority of them reported average material status—54.6%, *n* = 100 (*p* < 0.001) (Table 1).

### 3.2. QOL Scores of Study Population (n = 193)

Analysis of overall perception of quality of life showed that 39.4% (*n* = 73) of CVD patients were very dissatisfied or 36.8%v (*n* = 68) were dissatisfied with their quality of life. In addition, 34.1% (*n* = 63) were dissatisfied and 24.3% (*n* = 45) were very dissatisfied with their health status. The results of the respondents in each area of quality of life are presented below. It was found that most respondents were satisfied with their functioning in the environmental domain—67% (*n* = 126) were somewhat satisfied and 27.7% (*n* = 52) were satisfied with their quality of life (QoL) in this domain. More than half of respondents rated themselves as somewhat satisfied with their QoL in the domains: psychological (59.6%, *n* = 112), physical (54.8%, *n* = 103), and social (53.7%, *n* = 101). Respondents rated their functioning lowest in the physical domain, with 26.6% (*n* = 50) somewhat dissatisfied and 1.6% (*n* = 3) dissatisfied, and in the psychological domain, 23.4% (*n* = 44) were somewhat dissatisfied and 1.1% (*n* = 2) dissatisfied (Table 2).

The analysis of the variables related to the perception of QoL and health status showed that the respondents were more satisfied with their health status than with their QoL (M—2.88 vs. M—2.21; *p* < 0.001). The assessment of the QoL in all four domains (physical, psychological, social relations, and environmental) showed that CVD patients rated their social domain higher than the physical one (M—13.02 vs. M—11.78; *p* < 0.001). Similarly, some significant differences were found between physical and environmental domains. Subjects rated their environmental domain higher than the physical domain more often (M—13.26 vs. M—11.78; *p* < 0.001). The analysis also revealed significant differences between variables such as psychological and social relations domains. Respondents ranked the social relations domain higher than the psychological domain more often (M—13.02 vs. M—11.81; *p* < 0.001). In addition, significant differences were found between the psychological and environmental domains. CVD patients rated the environmental domain higher than the psychological domain (M—13.26 vs. M—11.81; *p* < 0.001). No significant differences were found between physical and psychological domains or between social relationships and environmental domains (Table 3).

### 3.3. Patients’ Expectations of a PHC Physician/Nurse

Analysis of expectations of a PHC physician revealed that 60.6% (*n* = 117) of patients expected physicians to be more available when needed, and 63.2% (*n* = 122) did not expect to be informed about the course of the disease (*p* < 0.001). It was also found that more than half of the respondents (53.9%, *n* = 104) expected courtesy and understanding or interest (*p* = 0.28). Additionally, the majority (80.8%, n = 156) did not expect a physician to ensure privacy during the examination (Table 4).

When analyzing expectations of a PHC nurse, it was found that the majority of patients (85%, *n* = 164) did not expect higher manual skills in providing nursing care (*p* < 0.001). However, 54.9% (*n* = 106) expected wider availability in case of an emergency (*p* = 0.171). It was also observed that 46.1% (*n* = 89) of respondents expected courtesy or understanding and interest (*p* = 0.28). The majority of patients (82.9%, *n* = 160) did not expect to be given privacy during nursing activities (*p* < 0.001) (Table 4).

The majority of patients (71.5%, *n* = 133) reported that the awareness of a nurse’s presence and their home visits affected their sense of safety during treatment (*p* < 0.001). In addition, 66.5% (*n* = 121) of them reported an improvement in their psychological state after the nursing visits (*p* < 0.001). After the visits, they frequently reported improvement in their mood (39.4%, *n* = 76) (Table 4).

### 3.4. The Correlation between CVD Patients’ Expectations of a PHC Physician and the Level of the QoL

The analysis of the correlations between the fulfillment of expectations and QoL revealed that the patients who did not expect the physician to inform them about the course of the disease rated the domains: environmental (r = 0.20, *p* = 0.006), psychological (r = 0.18, *p* = 0.015), physical (r = 0.18, *p* = 0.013), and social relations (r = 0.16, *p* = 0.025) significantly worse than those who expected to be informed. In addition, respondents who did not expect physicians to maintain privacy during examinations were more satisfied with their health status (r = −0.32, *p* < 0.001), QoL (r = −0.18, *p* = 0.014) and had lower levels of their QoL in physical (r = 0.27, *p* < 0.001), environmental (r = 0.23, *p* < 0.001), psychological (r = 0.21, *p* = 0.003), and social relations (r = 0.14, *p* = 0.048) domains compared to those who expected physicians to maintain privacy (Table 5).

### 3.5. The Correlation between Expectations of a Family Nurse and the Level of the QoL

The analysis of the correlation between CVD patients’ expectations of a family nurse practitioner and the level of QoL showed that respondents who did not expect a family nurse practitioner to be courteous and show understanding or interest rated the scores of the psychological (r = −0.17, *p* = 0.023) and physical (r = −0.15, *p* = 0.044) domains of the QoL higher than those who expected such behaviors. It was also observed that those who did not expect to be offered privacy during nursing activities rated higher levels of satisfaction with their health status (r = −0.15, *p* = 0.038) and the QoL (r = −0.14, *p* = 0.045) and lower scores in physical (r = 0.21, *p* = 0.004), social relations (r = 0.19, *p* = 0.010), and psychological (r = 0.16, *p* = 0.024) domains compared to those who expected to be offered it (Table 6).

### 3.6. The Correlation between CVD Patients’ Sense of Safety (Awareness of a Family Nurse’s Presence and Her Home Visits) and Improvement in Psychological Well-Being after Home Visits vs. Level of QoL

The correlation between CVD patients’ sense of safety (awareness of the presence of a family nurse and their visits) and improvement in their psychological well-being and level of QoL was also the subject of in-depth analysis. It was found that respondents who reported improvement in their psychological well-being and sense of safety after a family nurse visit improved their physical (respectively: r = 0.28, *p* < 0.001; r = 0.23, *p* = 0.001), psychological (r = 0.27, *p* < 0.001; r = 0.19, *p* = 0.010), and environmental domains of the QoL (r = 0.21, *p* = 0.004; r = 0.16, *p* = 0.031, respectively), which were much lower than those who reported no changes in the above aspects or had no opinion on them. On the contrary, those who did not notice positive changes in mood after the visits had higher scores in the environmental domain (r = −0.15, *p* = 0.043) than those who did.

## 4. Discussion

Quality of life assessment of CVD patients is a very important indicator of disease management and successful therapy [2]. Cardiovascular disease is most associated with lower quality of life [4,22,23,24], and the results of the study are consistent with these findings. It was found that most of the population in the study were dissatisfied with their quality of life and health status. In addition, respondents were more satisfied with their health status than with their quality of life (36.6%, *n* = 67 versus 21%, *n* = 39). When the scores were analyzed, it was found that CVD patients rated their physical and psychological functioning the worst. This is consistent with the argument that the nature of the disease can cause both physical [24] and psychological limitations [25,26]. It is worth noting that the median duration of CVD in the study group was 10 years. Other studies investigating the influence of selected medical factors on the quality of life of CVD patients showed that the worst physical and psychological outcomes were obtained in the patients whose diseases lasted longer [22]. The study by Sawicka et al. (2016) provides evidence that the duration of the disease affects the quality of life. They found that the shorter the course of the disease, the higher the quality of life [27].

The degree of patients’ satisfaction correlates positively with the improvement of their clinical condition. It has been proven that patient-centered health care (patient-centered care) is associated with a lower risk of death or hospital complications [15]. Therefore, better recognition of patients’ expectations of the health care system leads to higher satisfaction with visits and increases the chance of clinical improvement noted at follow-up visits, as well as their sense of safety [15,19,28]. It was found that 98% of patients in primary care have at least one expectation before visiting a physician [15]. The self-report revealed that 60.6% of respondents expected greater availability of the PHC physician when needed and 53.95% expected courtesy or understanding and interest. Another important issue is communication. Good communication between patient and physician is more meaningful than a plan and a course of health care. Patients who felt ignored or disoriented showed lower levels of satisfaction, as appropriate communication promotes overall patient well-being. Explaining the cause of illness, the goals of tests, and their results contributes to health-promoting behaviors. In addition, information about the course of the disease and its treatment, as well as its severity, helps to follow the physician’s recommendations [15,19]. Self-report revealed that only 36.8% of patients expected the physician to inform them about the course of the disease, but more than half (53.9%) expected courtesy and understanding or interest. Lack of expectations in this regard was also shown to correlate with quality of life. The patients who did not expect information from the physician reported lower scores in all QoL domains. It is also worth noting that the improvement in follow-up examinations was seen in these patients who made more effort to obtain information and were emotionally engaged during visits to the physician [29]. It is important to emphasize that explanations about the cause of the disease, its goals and test results, and the course of further treatment, together with explanations about how severe the disease may be, are an indispensable element in building a health care model based on patient co-decision [15]. The self-reported study shows that such measures need to be encouraged as part of a primary health care approach for CVD patients. An association was found between quality of life and lack of expectation of privacy during physical examinations. Respondents who had no expectation of privacy recorded higher scores on quality of life and their health status, but lower scores on QoL domains such as physical, psychological, social, or environmental. The topic receives little attention in scientific papers, and it is difficult to find appropriate comparative conclusions. However, it would be interesting to increase the research group to obtain more valuable results. However, the study proves that the problem exists and is worth studying and researching.

A nurse is also an important health care professional with whom a patient frequently comes into contact. Caring for a patient is a central theme of nursing. These are nurses who can make independent and competent decisions, assume personal and professional responsibilities, collaborate with the patient, their families, and interdisciplinary therapeutic teams to provide quality care, and maintain the patient’s health at the highest possible level. Therefore, assessing CVD patients’ expectations of their family nurses seems to be an important indicator of health care effectiveness. Kapała and Skrobisz (2006) concluded that nursing staff are expected to have manual skills and adequate information flow [30]. The self-reported study found that most CVD patients (85%, *n* = 164) did not expect higher manual skills in the delivery of care (*p* < 0.001) and therefore further research in home care is needed. Another study found that 70% of patients expected nurses to provide understanding and support regarding their illness or condition. Up to 92% of patients expected kindness, courtesy, patience, and caring from a PHC nurse [18]. It is worth noting that in self-report, 54.9% of patients expected greater availability when needed and 46.1% expected courtesy and understanding or interest. It should be mentioned that expectations of politeness, understanding, and interest correlated with lower scores in the psychological and physical domains of quality of life. The results definitely call for further research on this topic. It was also reported that lack of expectations regarding privacy during care correlated with higher scores for satisfaction with one’s health status and quality of life and lower scores in the physical, psychological, and social domains of QoL. At this point, it should be emphasized that ensuring privacy during care is a key issue. In the study by Wołosiewicz et al. (2013), 95% of respondents expected privacy and dignity to be respected during such interventions [18]. Improving feelings of safety during treatment and improving psychological well-being through regular visits and awareness of a nurse’s presence was correlated with lower scores in physical, psychological, and environmental domains of QoL. These characteristics define a target population that should be the target of programs to support home care of CVD patients by family nurses in PHC.

It should be noted that patients’ evaluation of medical services is a fundamental element for improving the quality of health care services and collecting information about the development guidelines and health needs of specific patient groups [15]. Addressing the challenges related to CVD in primary health care requires the development of health care quality assessment tools and the analysis of patients’ somatic, psychological, social, and environmental needs and expectations. Patients who fit the above characteristics should be targeted with medical and social programs that help maintain their health status and improve their QoL. Radical changes should be made in areas such as informing patients about the disease process, providing emotional support, paying attention to social and environmental functioning (including patients’ material status), and stimulating health-promoting behaviors that facilitate self-care and enhance quality of life.

### Strengths and Limitations of the Study

In summary, the limitations of the study may be the size of the study sample, which significantly limits the ability to interpret the results for the entire population of CVD patients in Poland. However, the results are valuable and can be used as supportive interventions for the development of a systemic care model for CVD patients who remain in home care. Accordingly, we suggest more extensive studies involving a larger number of patients and institutions, as well as broader application of quality of life assessment in patients with CVD. Future research should consider: the relationship between religiosity and spirituality, anxiety and depression, and quality of life in patients with CVD [31,32], as well as a range of other factors associated with quality of life in this patient population.

## 5. Conclusions

In our study, we found that there is a need to define the expectations of patients with chronic cardiovascular disease in the primary care setting, because the lack of expectations of a physician/nurse continues to be associated with lower quality of life in all quality of life domains.

## Figures and Tables

**Table 1 ijerph-19-03300-t001:** Sociodemographic data of patients with CVD (*n* = 193 *).

Variable	*n*	M	SD	Q. 25%	Me	Q. 75%	Min	Max	SW Test *p*
Age (in years)	193	70.50	16.90	62.00	74.00	84.00	18.00	100.00	*p* < 0.001
Duration of illness CVD	187 *	11.44	7.95	5.00	10.00	15.00	1.00	36.00	*p* < 0.001
**Variable**	**Categories**				** *n* **	**%**	**Chi^2^** **Test**		
Gender	Women				131	68.2	χ^2^—25.52 df—1 *p* < 0.001		
	Men				61	31.8			
	Total				192 *	100			
Education	Primary				59	31.2	χ^2^—128.3 df—6 *p* < 0.001		
	Vocational				48	25.4			
	Secondary without Matura exam				48	25.4			
	Secondary with Matura exam				10	5.3			
	Postsecondary				4	2.1			
	BA				19	10.1			
	MA				1	0.5			
	Total				189 *	100			
Staying in a relationship	Yes				88	47.3	χ^2^—0.54 df—1 *p* = 0.463		
	No				98	52.7			
	Total				186	100			
Place of residence	Urban				117	60.6	χ^2^—8.71 df—1 *p* = 0.003		
	Rural				76	39.4			
	Total				193	100			
Financial situation	Very good (above PLN 3001 per person in the family)				5	2.7	χ^2^—175.88 df—4 *p* < 0.001		
	Good (from PLN 2001–3000 per person in family)				47	25.7			
	Average (from PLN 1001–2000 per person in family)				100	54.6			
	Bad (from PLN 501–1000 per person in family)				30	16.4			
	Very bad (up to PLN 500 per person in family)				1	0.5			
	Total				183 *	100			

Legend: n—group quantity; %—percentage; M—mean; SD—standard deviation; Q. 25%—first quartile; Me—median; Q. 75%—third quartile; Min.—minimum; Max.—maximum; *p*—calculated level of significance for standard test Shapiro–Wilk; BA—bachelor’s degree; MA—Master’s degree; χ^2^—test statistic Chi^2^; df—degrees of freedom * The figures in column *n* do not sum up to 193 due to missing data.

**Table 2 ijerph-19-03300-t002:** The distribution of WHOQOL-BREF variables in the research group (*n* = 193 *).

Variable	Categories		*n*	%
	Number of Points on the Scale	Evaluation		
WHOQOL-BREFQoL Perception	1	Very dissatisfied	73	39.5
	2	Dissatisfied	68	36.8
	3	Neither satisfied nor dissatisfied	5	2.7
	4	Satisfied	11	5.9
	5	Very satisfied	28	15.1
	Total		185 *	100.0
WHOQOL-BREFHealth Perception	1	Very dissatisfied	45	24.3
	2	Dissatisfied	63	34.1
	3	Neither satisfied nor dissatisfied	10	5.4
	4	Satisfied	4	2.2
	5	Very satisfied	63	34.1
	Total		185 *	100.0
WHOQOL-BREFPhysical Domain	(0–5)	Dissatisfied	3	1.6
	(5–10)	Rather dissatisfied	50	26.6
	(10–15)	Rather satisfied	103	54.8
	(15–20)	Satisfied	32	17.0
	Total		188 *	100.0
WHOQOL-BREFPsychological Domain	(0–5)	Dissatisfied	2	1.1
	(5–10)	Rather dissatisfied	44	23.4
	(10–15)	Rather satisfied	112	59.6
	(15–20)	Satisfied	30	16.0
	Total		188 *	100.0
WHOQOL-BREFSocial Relations Domain	(0–5)	Dissatisfied	1	0.5
	(5–10)	Rather dissatisfied	35	18.6
	(10–15)	Rather satisfied	101	53.7
	(15–20)	Satisfied	51	27.1
	Total		188 *	100.0
WHOQOL-BREFEnvironmental Domain	(0–5)	Dissatisfied	1	0.5
	(5–10)	Rather dissatisfied	9	4.8
	(10–15)	Rather satisfied	126	67.0
	(15–20)	Satisfied	52	27.7
	Total		188 *	100.0

Legend: n—group quantity; %—percentage. * The figures in column *n* do not sum up to 193 due to missing data.

**Table 3 ijerph-19-03300-t003:** The assessment of QoL according to the WHOQOL-BREF (*n* = 193 *).

Variable	*n*	M	SD	Min	Q. 25%	Me	Q. 75%	Max	Wilcoxon Test		Quade Test			
									V	*p*				
WHOQOL-BREF QoL Perception	185 *	2.21	1.41	1	1	2	2	5	898	**0**				
WHOQOL-BREF Health Perception	185 *	2.88	1.64	1	2	2	5	5						
WHOQOL-BREF Physical Domain	188 *	11.78	3.39	4.57	9.71	12	13.86	19.43			* **p** *	**PD**	**PsD**	**SRD**
WHOQOL-BREF Psychological Domain	188 *	11.81	2.93	4	10	11.33	14	19.33			**PsD**	0.81	-	-
WHOQOL-BREF Social Relations Domain	188 *	13.02	3.24	4	10.67	13.33	16	20			**SRD**	**0**	**0**	-
WHOQOL-BREF Environmental Domain	188 *	13.26	2.47	4.5	11.5	13.5	15	19.43			**ED**	**0**	**0**	0.65

Legend: PD—WHOQOL-BREF physical domain; PsD—WHOQOL-BREF psychological domain; SRD—social relationship domain; ED—environmental domain; n—group quantity; M—mean; SD—standard deviation; Q. 25%—first quartile; Me—median; Q. 75%—third quartile; Min.—minimum; Max.—maximum; Wilcoxon test: V—value of test statistic; *p*—calculated significance level; Quade test of multiple comparisons: *p*—calculated significance level of the Quade test for each pair of variables (domains). Calculated significance levels of p which value was less than 0.05 are highlighted in bold. * The figures in column *n* do not sum up to 193 due to missing data.

**Table 4 ijerph-19-03300-t004:** The expectations of CVD patients towards a PHC physician or nurse (*n* = 193 *).

Variable	Categories	*n*	%	Chi^2^ Test
Expectations towards a PHC physician	None of the below	76	39.4	χ^2^—8.71 df—1 *p* = 0.003
	Higher availability in case of a necessity	117	60.6	
	Total	193	100	
	None of the below	122	63.2	χ^2^—13.48 df—1 *p* < 0.001
	Informing about course of disease	71	36.8	
	Total	193	100	
	None of the below	89	46.1	χ^2^—1.17 df—1 *p* = 0.028
	Courtesy, showing understanding and interest	104	53.9	
	Total	193	100	
	None of the below	156	80.8	χ^2^—73.37 df—1 *p* < 0.001
	Providing privacy during examinations	37	19.2	
	Total	193	100	
Expectations towards a family nurse	None of the below	164	85	χ^2^—94.43 df—1 *p* < 0.001
	Higher manual skills in nursing activities	29	15	
	Total	193	100	
	None of the below	87	45.1	χ^2^—1.87 df—1 *p* = 0.171
	Higher availability in case of a necessity	106	54.9	
	Total	193	100	
	None of the below	104	53.9	χ^2^—1.17 df—1 *p* = 0.28
	Courtesy, showing understanding and interest	89	46.1	
	Total	193	100	
	None of the below	160	82.9	χ^2^—83.57 df—1 *p* < 0.001
	Providing privacy in nursing activities	33	17.1	
	Total	193	100	
Does the consciousness of the nurse’s presence and their visits make you feel safer in the course of your disease?	Yes	133	71.5	χ^2^—134.23 df—2 *p* < 0.001
	No	7	3.8	
	No opinion	46	24.7	
	Total	186 *	100	
Do the nursing visits improve your mental well-being?	Yes	121	66.5	χ^2^—103.86 df—2 *p* < 0.001
	No	10	5.5	
	No opinion	51	28	
	Total	182 *	100	
If “yes”, why?	None of the below	117	60.6	χ^2^—8.71 df—1 *p* = 0.003
	I always feel better after a visit	76	39.4	
	Total	193	100	
	None of the below	144	74.6	χ^2^—46.76 df—1 *p* < 0.001
	I have more strength to fight the disease and its symptoms	49	25.4	
	Total	193	100	
	None of the below	162	83.9	χ^2^—88.92 df—1 *p* < 0.001
	I am full of hope and strength	31	16.1	
	Total	193	100	

Legend: n—group quantity; %—percentage; χ^2^—test statistic Chi^2^; df—degrees of freedom. * The figures in column *n* do not sum up to 193 due to missing data.

**Table 5 ijerph-19-03300-t005:** The correlation between expectations of a PHC physician and the level of the QoL.

Variable	WHOQOL-BREF											
	QoL Perception		Health Perception		Physical Domain		Psychological Domain		Social Relations Domain		Environmental Domain	
What are your expectations towards a PHC physician?												
	r	*p*	r	*p*	r	*p*	r	*p*	r	*p*	r	*p*
Higher availability in case of a necessity	−0.07	0.351	−0.04	0.606	0.06	0.438	0.03	0.674	0.08	0.286	−0.01	0.868
Informing about the course of the disease	−0.04	0.538	−0.13	0.068	0.18	0.013	0.18	0.015	0.16	0.025	0.20	0.006
Courtesy, showing understanding and interest	0.07	0.330	−0.08	0.243	0.07	0.363	0.07	0.324	0.05	0.462	0.12	0.092
Providing privacy during examinations	−0.18	0.014	−0.32	<0.001	0.27	<0.001	0.21	0.003	0.14	0.048	0.23	0.002

Legend: r—Spearman’s correlation coefficient, r—at *p* ≤ 0.05.

**Table 6 ijerph-19-03300-t006:** The correlation between the expectations towards a family nurse and the level of the QoL.

	WHOQOL-BREF											
Variable	QoL Perception		Health Perception		Physical Domain		Psychological Domain		Social Relations Domain		Environmental Domain	
What are your expectations towards a family nurse?												
	r	*p*	r	*p*	r	*p*	r	*p*	r	*p*	r	*p*
Higher manual skills during nursing activities	−0.01	0.871	−0.10	0.154	0.04	0.610	−0.02	0.792	0.01	0.916	−0.07	0.361
Higher availability in case of a necessity	0.00	0.974	−0.06	0.375	0.07	0.370	0.08	0.251	0.02	0.771	0.00	0.962
Courtesy, showing understanding and interest	0.12	0.109	0.05	0.468	−0.15	0.044	−0.17	0.023	−0.03	0.643	−0.09	0.229
Providing privacy during nursing activities	−0.14	0.045	−0.15	0.038	0.21	0.004	0.16	0.024	0.19	0.010	0.13	0.067

Legend: r—Spearman’s correlation coefficient, r—at *p* ≤ 0.05.

## Data Availability

The data presented in this study are available on request from the corresponding author.

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
