# Peer review of "The Analysis of the Relationship between the Quality of Life Level and Expectations of Patients with Cardiovascular Diseases under the Home Care of Primary Care Nurses"

_ijerph, 2022, doi:10.3390/ijerph19063300_

Round 1
Reviewer 1 Report
Szlenk-Czyczerska et al conducted a study to investigate the relationship between the quality of life and expectations of patients with cardiovascular diseases. This is interesting area of research, however, there are several concerns regarding the study methodology and the way the findings have been presented in this manuscript.
- It is not clear from the abstract that a) Besides having the chronic disease condition, what are the other factors affecting the CVD patients' level of quality of life?
- b) whether the "expectations of patients being met" refers to quality of treatment, medical insurance or financial support, or social-mental-environmental support?
- Abstract: Provide details on study participants, setting and duration of the study
- what is “Author’s interview Questionnaire”? is it a validated questionnaire? what is WHOQOL-BREF questionnaire? was it specifically designed for this study?
- Explicitly mention the rationale for using Spearman correlation coefficient?
- abstract, line 26: what do you mean by "quality of QoL"?
- Introduction: line 64, “it was reported that lower QoL indicators were found in patients taking advantage of health care services more often”. What are those lower QoL indicators? briefly discuss.
- Materials and methods, line 104-106: "CVD patients under the home care family nurses", it should be reflected in the title.
- Sample, lines 111-114. were these questionnaires administered by the nurses or an independent interviewer who did not complete an additional questionnaire concerning the patient?
- One of the criteria for inclusion in the study was 18 year of age or above. However, according to Table 1, the minimum age of the CVD patients was 17 year.
- Variables and data collection: Include the study questionnaires in the supplemental material.
- Data analysis: It is not clear why authors have mentioned “Publisher, City, Postal code, Country” in page 4, line 151.
- The rationale for the use of the Wilcoxon's test and Quade test is not well described. It was mentioned in the abstract that “The data analysis was based on the Spearman correlation coefficient test”; however, I do not see those methods in the data analysis section.
- Table 1: why were there so many categories for education given that the number of participants in each category is limited? what is Matura exam? what is the relevance of categorizing secondary education with and without Matura exam?
- Financial situation: how was this variable defined? the response is very subjective, and the perception or understanding of financial situation could vary between the study participants.
- Table 2 is not informative. Describe the questions (1 and 2) and items within each domain (for example, physical domain, psychological domain, etc). how many items were included in each domain? Clarify the grouping of number of points on the scale for each domain.
- Too many abbreviations; hard to follow and interpret the findings of the Tables 3-6, with little or almost no information about the individual response items/domains.
Author Response
Drodzy Redaktorzy
We would like to sincerely thank the Editorial Board and the Reviewer of your esteemed International Journal of Environmental Research and Public Health for their positive feedback and constructive recommendations to improve our observational paper entitled „The Analysis of the Relationship Between the Quality of Life Level and Expectations of Patients with Cardiovascular Diseases Under the Home Care of Primary Care Nurses” by Elżbieta Szlenk-Czyczerska, Marika Guzek, Dorota Emilia Bielska, Anna Ławnik, Piotr Polański and Donata Kurpas.
In this first round of review, we focused heavily on the points raised in your letter. We would like to respond to this statement point by point based on our careful revision, as you can see in the table below. Accordingly, the final version of the manuscript text includes all necessary changes and improvements.

Reviewer 2 Report
This paper addresses an important issue regarding the quality of life of patients with chronic diseases. It emphasizes the need to consider quality of life for functional well-being evaluation and treatment planning. The research is well conducted and results are very interesting and informative. There are some points of concern intended for further improvement of the quality of the study and more suggestions.
The manuscript needs extensive editing by a native English speaker to amend all the errors.
Title needs some editing and language check: Their should be removed
The Analysis of the Relationship Between the Quality their of Life Level and the Expectations of Patients with Cardiovascular
Many other examples like
and social: r = 0.16, p = 0.025. In patients who did not expect the nurse to be courteous, show under- line 28
Worldwide burden is connect
ed with cardiovascular diseases, an anticipated increase line 80
The study was conducted according to the we non-probabilistic sampling method line 116
expectations regarding intimacy during nursing activities and the level of satisfaction with one's….. on line 31, should be upon being attended for by nurses….
-Abbreviations should be properly introduced throughout the text.
-In the Methodology, the researchers mention that :” This study is part of a broader study….. to identify indicators that determine the effectiveness of home care for patients with chronic….” line 98, can the authors elaborate how overlap would be avoided as it is stated clearly there is a partial analysis of the data….
-I suggest to include a simple diagram explaining recruitment sites and sample size
- The implications summed at the end of the manuscript are very important, I suggest however to add an aspect for future researchers to elaborate on, for instance consider correlation mechanisms to cope, also adding more recent references.
- Abu HO, Ulbricht C, Ding E, Allison JJ, Salmoirago-Blotcher E, Goldberg RJ, Kiefe CI. Association of religiosity and spirituality with quality of life in patients with cardiovascular disease: a systematic review. Qual Life Res. 2018 Nov;27(11):2777-2797. doi: 10.1007/s11136-018-1906-4. Epub 2018 Jun 11. PMID: 29948601; PMCID: PMC6196107.
- Jha MK, Qamar A, Vaduganathan M, Charney DS, Murrough JW. Screening and Management of Depression in Patients With Cardiovascular Disease: JACC State-of-the-Art Review. J Am Coll Cardiol. 2019 Apr 16;73(14):1827-1845. doi: 10.1016/j.jacc.2019.01.041. PMID: 30975301; PMCID: PMC7871437.
Author Response
Dear Editor,
We would like to sincerely thank the Editorial Board and the Reviewer of your esteemed International Journal of Environmental Research and Public Health for their positive feedback and constructive recommendations to improve our observational paper entitled „The Analysis of the Relationship Between the Quality of Life Level and Expectations of Patients with Cardiovascular Diseases Under the Home Care of Primary Care Nurses” by Elżbieta Szlenk-Czyczerska, Marika Guzek, Dorota Emilia Bielska, Anna Ławnik, Piotr Polański and Donata Kurpas.
In this first round of review, we focused heavily on the points raised in your letter. We would like to respond to this statement point by point based on our careful revision, as you can see in the table below. Accordingly, the final version of the manuscript text includes all necessary changes and improvements.

Round 2
Reviewer 1 Report
Thank you for the revisions.